# Revisiting Multifactor Models of Dyslexia: Do They Fit Empirical Data and What Are Their Implications for Intervention?

**DOI:** 10.3390/brainsci13020328

**Published:** 2023-02-14

**Authors:** Maria Luisa Lorusso, Alessio Toraldo

**Affiliations:** 1Scientific Institute IRCCS E. Medea, 23842 Bosisio Parini, Italy; 2Department of Brain and Behavioral Sciences, University of Pavia, 27100 Pavia, Italy; 3Milan Center for Neuroscience, NeuroMI, 20126 Milan, Italy

**Keywords:** developmental dyslexia, multifactor-interactive model, multiple-deficit model, single-deficit model, subtypes, intervention, balance model, inter-hemispheric integration, hemisphere-specific stimulation

## Abstract

Developmental dyslexia can be viewed as the result of the effects of single deficits or multiple deficits. This study presents a test of the applicability of a multifactor-interactive model (MFi-M) with a preliminary set of five variables corresponding to different neuropsychological functions involved in the reading process. The model has been tested on a sample of 55 school-age children with developmental dyslexia. The results show that the data fit a model in which each variable contributes to the reading ability in a non-additive but rather interactive way. These findings constitute a preliminary validation of the plausibility of the MFi-M, and encourage further research to add relevant factors and specify their relative weights. It is further discussed how subtype-based intervention approaches can be a suitable and advantageous framework for clinical intervention in a MFi-M perspective.

## 1. Introduction

According to the ICD-10 criteria [1], developmental dyslexia (DD) is diagnosed in children who fail to develop normal reading skills in spite of normal intelligence, adequate motivation and schooling. 

While reading difficulties are the hint that most often leads to a clinical diagnosis, they probably constitute only the most visible dysfunction of a more composite picture of a defective neurocognitive profile, which is likely to extend across different functions [2]. Although phonological and metaphonological deficits have been described as the most reliable and specific distinctive features of dyslexic readers (e.g., [3]), several other functions were shown to be specifically impaired in dyslexia, including long-term and short-term verbal memory or working memory [4,5,6,7], visual and auditory perception [8,9,10] and visual attention [11,12,13]. 

### 1.1. Single- and Multiple-Deficit Models of DD

There are three possible explanations for this heterogeneous pattern of findings: (a) there is only one core deficit causally related to the reading impairment, while the other deficits are just associated disorders without any direct causal relationship with reading; (b) different factors constitute the core deficit in different subtypes of dyslexia; (c) the different deficits are different manifestations of the impairment of a superordinate system—thus, they are functionally related to each other. Both (a) and some formulations of (b) may be considered as *single-deficit* models of dyslexia, insofar as they posit that one deficit (even if not always the same) is responsible for the reading difficulty in each individual. Additionally, (c) might be classified as a single-deficit model if the system proposed to be responsible of the reading problem is a very restricted system of closely related functions; e.g., phonological discrimination and phonological representation—and maybe phonological retrieval.

Indeed, the most widely accepted approach posits that developmental dyslexia arises from deficits in systems that are linguistic in nature [14]. In particular, the “phonological deficit” theory suggests that DD is caused by a deficit in phonological processing and memory (e.g., [3,15,16,17]), affecting the development of phonological representations that are the foundation of orthographic development. This theory is supported by correlational evidence linking the development of phonemic awareness to the development of literacy skills, intervention studies showing better reading after training of phonological skills [18,19,20], and neuroimaging studies showing poor activation in left posterior areas in dyslexic individuals (e.g., [21]). 

A different framework points to deficits in the visual processes mediated by the magnocellular system [8,22,23]; i.e., the part of the visual system that is usually identified with the “transient” subsystem (as opposed to the “sustained” subsystem). A deficit in callosal inter-hemispheric processing as a possible cause of dyslexia has been suggested by other authors [24,25,26]. Korkmaz and colleagues [27] found that among adult healthy readers, slow readers have a slower right-to-left callosal transfer time at the parietal level, which is related to the visual word decoding process. Right-to-left callosal transfer is essential for the integration of text coming from the left visual field and that coming from the right visual field at the appropriate time [28]. 

In the past two decades it has thus become more and more evident that single-deficit or core-deficit models of dyslexia cannot explain its complexity and heterogeneity. For this reason, new views of dyslexia have been proposed that take into consideration multiple sources of impairment and that are generally indicated with the label of *multiple-deficit* models (MDM) of dyslexia [29]. These models have been received with mixed reactions, from overt subscriptions [30,31,32,33,34,35] to attempts to reconcile them with core-deficit models (e.g., [36,37]), to complete neglect of this theoretical framework. MDMs posit that dyslexia results from the impairment of a series of functions, and that these functions need not always be the same ones in all cases. As thoroughly explained in one of the seminal papers proposing MDM approaches [38], comparing two types of single-deficit, two types of multiple deficits, and a hybrid model where both single deficits and multiple deficits are allowed, it is the hybrid model that shows the best fit in a sample of real cases. This kind of model, completely overcoming a categorical approach in favor of a dimensional and probabilistic one is exactly what would be expected in the perspective that will be described below, which considers many different functions that are known to have a role in reading and thus can logically be expected to have a potential role in reading disorders. While the authors discuss possible implications for diagnostic purposes, it should be considered that diagnosis of DD is and should be a behavioral one, based on actual reading and spelling performance—there would be no advantage in introducing a different diagnostic approach based on neuropsychological or neurobiological indicators. Nonetheless, as also suggested in the paper mentioned above, the availability of such indicators might be advantageous in terms of early detection of at-risk situations and prevention in the form of empowerment of the targeted functions. Moreover, this kind of information could be extremely relevant for intervention. Indeed, in the context of a single-deficit model of DD, intervention should address the function that is supposed to constitute the “core” of the problem. The predominance of phonology- and phonological awareness-based intervention programs, especially in the US, is clearly consequent to the predominance of phonological-deficit models of DD (not disputing their effectiveness, often experimentally proven, but just observing the scarcity of alternative programs). A logical consequence of the adoption of a MDM of DD would be the use of intervention models addressing multiple functions. Nonetheless, very few validated programs exist which adopt this perspective, and the alternative to phonology-based programs mainly consists in programs tapping either visual functions or attentional functions in isolation (a choice that often derives from the need to demonstrate the role of a given mechanism in the etiology of DD; i.e., once more a single-deficit approach). A notable exception could be seen in Tallal’s Fast ForWord program, whose effectiveness, however, has not been supported by recent meta-analyses [39], and few other programs. For instance, [40]’s meta-analysis subdivides intervention programs for DD into six categories: phonemic awareness, phonics, reading fluency, reading comprehension and auditory training, medical treatment and colored overlays: none of the studies was a multifactor treatment. The authors of [41] in their recent meta-analysis state that 35 of the 55 included studies were multicomponential, but multicomponentiality in this case indicated the combination of various types of language-related functions (e.g., phonics and reading fluency, or phonological awareness and reading comprehension) and did not span over non-linguistic functions. In [42]’s review, the 40 selected studies included new approaches such as brain stimulation, action video games, visual/visual-attentional trainings, working memory training, modeling (a Bandura-inspired approach), reading acceleration (addressing eye movements), vergence training and multisensory stimulation approaches, but there were only three mixed treatments combining elements from multiple approaches. Therefore, even if the last meta-analysis shows that the “mixed approaches” were characterized by rather limited effect sizes, the heterogeneity and limited number of studies hardly constitute sufficient evidence to draw any conclusion about the effects of a multifactor approach to intervention. Neuroimaging studies of intervention effects, on the other hand, suggest that great individual variability likely masks changes in brain activation following treatment for DD [43], and that different intervention types may lead to similar improvements through different mechanisms and pathways [44].

### 1.2. Multiple-Deficit Models of DD as a Non-Unitary Concept

It should be acknowledged that under the label MDM one can find different and sometimes only partially overlapping views of DD, or even more simply, different approaches to its investigation. 

One line of research conceptualizes multiple factors as variables located at different levels: typically, genetic, neurobiological, cognitive, behavioral [36,45,46]. As suggested by [46], this approach should probably be defined as *multidimensional* rather than multifactorial, since it observes the mutual interactions of factors that are not at the same logical level, but rather span over various dimensions that can generally be ordered in terms of the directions of causal relationships. Thus, several studies have searched for risk and protective factors at the genetic and environmental levels that could modulate the expression of other genetic or environmental factors on the cognitive and behavioral levels [45,47,48,49,50]. Clearly, this approach is reasonable but also very ambitious, considering that the interactions of factors across levels are still poorly known and ill-defined. Still, this is likely the direction research on neurodevelopmental disorders needs to go in the future. The present proposal can be considered as an early step, trying to bring more clarity at just one of those levels, i.e., the cognitive-behavioral one.

Other studies have conceptualized multifactoriality as the presence of shared protective and risk factors at the basis of comorbidity [35,51,52]. Some of these studies [3,53] have applied cutoffs to the various measures in order to define sets of “impaired” vs. “non-impaired” functions, imposing a traditional categorical approach where a dimensional one might have been more appropriate. Indeed, the results of such studies give only partial responses to the issues raised by MDM theories needing to define the functions that may contribute to the manifestations of DD, irrespective of their severity (and possibly, irrespective of the severity of the reading deficit itself). Many other studies embrace a MDM view, but practically focus on only a very limited number of functions, sometimes just functions that are all related to phonological processing (e.g., [2]), thus taking an intermediate position between a MDM perspective and a core-deficit one. 

### 1.3. The Multifactor-Interactive Model (MFi-M) of DD

The present work suggests that the set of functions that cause the reading difficulty is much larger and heterogeneously composed than usually claimed. In principle, any function that has a role in reading can be (part of the) source of a reading difficulty. Classical neuropsychological models of reading describe the reading function as the output of two or three different sequences of steps involving *either* (i) character decoding and recognition, grapheme-to-phoneme conversion, phoneme assembly and (optionally) retrieval from the store of visual or phonological forms corresponding to a certain meaning, *or* (ii) search in the store of visual word forms until a corresponding unit is found that will be linked to a certain phonological form to be activated [54]. In the case of reading aloud, the final step in both cases would be the retrieval, activation and execution of an additional articulatory pattern and program. This description calls into play a certain number of functions, belonging to the visual, phonological and articulatory–motor domains. However, the number of subprocesses involved is much larger, since practically any of these steps involve other factors acting as subconstituent elements, and thus exerting a role in the resulting reading process. For instance, character recognition in the preliminary visual analysis of the letters (or syllables, or morphemes, or words) involves a large number of visual and attentional processes, from the coordination of eye movements to the modulation of the attentional focus to the control of crowding effects to the higher-level visual processing components requiring pattern identification, comparison, discrimination, recognition, etc. [55,56,57]. In a similar vein, the search for possible entries in the visual (or phonological) lexicon will probably be guided by many types of contextual and linguistic information (i.e., lexical, phonological, morphological, syntactic, semantic, pragmatic and general world-knowledge information) about the word to be retrieved, with a preactivation of subsets of words that are more likely to be the correct target [58,59,60]. Clearly, the relevance that each of these subcomponents may have in the whole decoding process can be extremely variable, as can be the consequences of a possible deficit, from just slightly slowing the process to reducing the probability of a correct retrieval. We may additionally reason that rather than the effect of a single process, it will be the interplay among all processes that determines the efficiency of the decoding process. In more quantitative terms, we may think of the decoding process as the *sum of a large number of processes* (many of which may be just optional ones), each one characterized by a specific “weight”. If the sum of these weights (which we shall call “MFi-index”, where MFi stands for *multifactor-interactive*) reaches a certain threshold, the process can take place and flow in a relatively efficient, flawless way; otherwise, there will be difficulties that will be proportional to the gap between the individual MFi-index and the threshold. The existence of such a threshold directly contradicts the notion of a simple additive model, since the impact of each component would not be constant, and independent of other components, but rather, it would change on grounds of the whole pattern of components. Namely, the impact of any given component would be small when other components are relatively spared/good, because their sum would be above the threshold for efficient processing; else, it would be amplified when the other components sum up at an efficiency level close to, or below the threshold. Therefore, the MFi-index should modulate the effect of each single component, which would be the larger, the lower the MFi-index. Another expected feature of the MFi-index is that it should be a fluid product changing constantly during the reading process itself, on the basis of the specific processes and microprocesses involved. Therefore, it would not be possible to identify a constant “sum” characterizing each reader, since this quantity will change continuously. Nonetheless, the same reader would very likely tend to have a certain average MFi-index across their reading experience, and this average would describe their general reading ability.

Why should the MFi-index be relevant to researchers and clinicians, and what would it add to the concept of multi-deficit models of dyslexia? 

Among aspects shared by the various MDMs, we can list the following: (i) MDMs posit that there are multiple possible risk factors for each neurodevelopmental disorder and that sharing some of these risk factors is the basis for comorbidity; (ii) MDMs also propose that each risk factor is probabilistically related to the occurrence of neurodevelopmental disorders. As claimed by some of its proponents [31], the challenges inherent in this kind of model are its almost non-falsifiability and its (present) incapacity to explain causality due to the absence of longitudinal studies. In their update of the theory, these authors wish for a greater degree of specification in the future, which may make the theory more falsifiable, and for longitudinal studies that may uncover developmental changes and trends in the ways the various deficits interact and give form to observable symptoms. These elements are very close to the multifactor model proposed here, and it could also be claimed that the latter provides some of the greater specification and quantitative definition that is solicited by [31] in order to make the theory testable. Nonetheless, the focus of MDMs is different from that of the present proposal in that they especially concentrate on comorbidity among different neurodevelopmental disorders; i.e., they are more focused on common deficits that may underlie different disorders than on deficits that may underlie a single disorder, DD in this case. Although the two goals are absolutely compatible and may actually be seen as two faces of the same coin, the practical consequence of pursuing the former or the latter goal is that more general neuropsychological functions (e.g., language, attention, memory, etc.) versus more specific microprocesses (e.g., attentional shifting, control of visual crowding, phoneme categorization or morphological analysis) may be the focus of the analysis in the two cases. In other terms, the direction could be more “horizontal”, looking for links and similarities in the first case, and more “vertical”, looking for micro-components and details, in the other case. 

Hence, because of its focus on microprocesses, the MFi-index is likely to introduce more variability and more flexibility into the concept of multi-deficit models of DD. Nonetheless, it could start providing a more quantitative model of the many factors involved in at least one of the various neurodevelopmental disorders that are so frequently comorbid, and apparently the most common one, dyslexia, and might give the opportunity for a first testing benchmark.

For instance, [31] view the presence of about 25% of children whose disorder can be accounted for by a single deficit as potentially problematic (even if they conclude it is not). By contrast, the present model would not be challenged by the finding that some children only have one weak function. Rather, it would predict that (reading performance being equal) the single deficit present in those children should, on the average, be larger than the average deficit of the same type observed in children who have two evident deficits, and this would be larger than the average deficit of the same type observed in children who have three apparent deficits, and so on. If the relevance of a certain deficit is, as it would seem plausible, proportional to the frequency with which it is found in the dyslexic population, then the influence, or weight, of a specific function should be proportional to the frequency with which that function is described as a “core deficit”. Following this line, one could expect that phonological awareness should have the highest starting weight, followed by lexical access (as measured by RAN—rapid automatized naming) and then, in a very tentative hypothesis based on previous work by [3,60], by visual processing (as measured by magnocellular tests) and/or visual-attentional skills (as measured by classical Posner visual focusing paradigms). Moreover, it could be anticipated that the same level of impairment in a given function could be found in an affected (dyslexic) child when the other relevant functions are also low or average, and in an unaffected (non-dyslexic) child when the other relevant functions are unimpaired (or even above the norm); i.e., they can act as protective factors balancing out the effect of the defective function. 

Nonetheless, the most relevant difference with respect to MDMs is probably to be seen on the clinical application level: a better specification of the constellation of functions with their specific weights, in fact, would provide an immediate feed for the definition of an individualized intervention program specifying the goals, the sequence (in terms of priority) and the strengths on which the program could capitalize.

In order to test the present model, we derived one simple empirical prediction. If it is true that the contribution by a single function to the emergence of dyslexia critically depends on the status of the other functions, we should find significant interactions between a single function and the “weighted sum” of the other functions. Thus, for instance, the effect produced by variation in short-term memory on the severity of dyslexia would be larger when other deficits are (collectively) large, and smaller when other deficits are (collectively) small. More specifically, it is anticipated that a model including, beyond a function and the sum of the remaining functions, also their interaction, will be more predictive of reading ability compared to the same model without the interaction (i.e., compared to a model in which the two variables only show additive effects). Although no empirical study was conducted to test the model (and related hypotheses) directly, its predictive capacity was preliminarily tested on an existing database of children with DD. 

## 2. Materials and Methods

### 2.1. Participants

For the present investigation, data from 55 children with developmental dyslexia, ranging in age between 7 and 13 years (mean age = 9.70 yrs., SD = 1.43), were selected from an existing database of 266 patients referred to the Institute and treated for DD. Even if the reading and writing tests were exactly the same for the whole sample, the number and type of non-reading tests administered in the various periods had changed due to specific research goals. Therefore, each single test or each combination of specific tests could be present in only a subgroup of children. The selection of the children to include for the present study was based on the availability of all the data needed to perform the analysis for the test of the MFi-M. Data from all the children who met these conditions were included in the sample and further analyzed.

All children in the database fulfilled the following inclusion criteria: (a) having been diagnosed with a specific reading disorder (ICD-10 codes: F81.0 or F81.3) on the basis of standard inclusion and exclusion criteria (ICD-10, [1]) (at least one z-score concerning reading/writing speed and/or accuracy below −2); (b) absence of comorbidity with other psychopathological conditions (whereas comorbidity with other learning disorders and/or ADHD was allowed); (c) not having been involved in other clinical intervention programs for learning disorders. The retrospective study was approved by the local ethics committee according to the declaration of Helsinki. All parents had signed informed consent.

### 2.2. Materials

The tests commonly employed in the assessment of specific learning disorders in Italy were used to assess reading. The results of the tests are expressed as z-scores according to age norms. The following tests were administered:Text reading: “*Prove di Rapidità e Correttezza nella Lettura del gruppo MT*” (“Test for Speed and Accuracy in Reading, developed by the MT group” [61,62]. This test assesses reading abilities for meaningful material. It provides separate scores for speed and accuracy. Texts increase in complexity with grade level. Norms are provided for each text. Validity and reliability for the MT text reading test are reported to be satisfactory without further specifications.Single word/nonword reading: “*DDE-2: Batteria per la Valutazione della Dislessia e Disortografia Evolutiva-2*” (Assessment Battery for Developmental Reading and Spelling Disorders-2; Refs. [63,64]). The battery assesses speed and accuracy (expressed in number of errors) in reading word lists (4 lists of 24 words) and nonword lists (3 lists of 16 nonwords) and provides grade norms from grade 2 of primary school to grade 3 of junior high school (corresponding to the 8th year of schooling). The DDE word and nonword reading test has an adequate reliability (mean test–retest coefficients are 0.77 for speed and 0.56 for accuracy).IQ tests: Cattell’s non-verbal, Culture Fair Test, Scale 2, Form A [65] was administered. The children whose scores were below 90 were further tested with the Wechsler Scale (the Italian versions of WISC-R [66] or WISC-III [67], depending on the year of testing), and non-verbal IQ was recorded. Only children whose non-verbal IQ was above 70 on the Wechsler scale were included in the database. For the purposes of the present study, IQ scores were converted into z-scores.

Moreover, a series of additional standardized neuropsychological tests had been administered. These tests included:Memory. *“Test di Memoria e Apprendimento*” (TEMA; Italian adaptation of TOMAL, Test of Memory and Learning; [68]) was used to assess short-term memory and working memory. Verbal short-term memory was assessed by means of immediate serial recall (span) of letters (forward letter span). Working memory was evaluated by backward recall of letters (backward letter span). The number of letters correctly recalled in the correct sequence was recorded. Age norms are provided. This test was administered to a total of 257 children.Rapid automatized naming: “*Denominazione Rapida*”—RAN [69]). Participants were shown 50 stimuli printed on paper, arranged in a 5 × 10 matrix, and were instructed to accurately name each stimulus as quickly as possible, in reading order (left to right, top to bottom). Two matrices were presented. One matrix contained color stimuli (squares: black, yellow, green, red and blue), another contained objects (figures: star, dog, pear, train and hand). The time needed to name all the elements of each matrix was measured by means of a stopwatch. Naming errors and omissions were also noted. Before the test, a practice trial was run with a smaller matrix (20 items). Two raw scores were recorded: speed, expressed in seconds, and accuracy, expressed in number of naming errors. Z-scores based on grade norms were calculated. This test was administered to a total of 55 children.Visual search. “*Ricerca Visiva*” (visual search—VS) [69] was used to assess speed and accuracy in visual search for familiar stimuli (figures, digits). Two matrices (10 rows of 5 stimuli each) for each kind of stimulus were presented and the child was requested to cancel one of the stimuli (the number “7” and the figure “star”) presented in the matrix as quickly and as accurately as possible. Two raw scores were recorded: speed, expressed in seconds, and accuracy, expressed in number of cancellation errors. Z-scores based on grade norms were calculated. This test was administered to a total of 55 children (the same who had undergone assessment with the RAN test).Phonological awareness: Two tests of phonemic awareness [70]. (i) Phonemic Elision: the test assesses the ability to recognize and isolate the phonemic constituents of 20 words (in this specific case, the initial constituents). The child was asked to delete the first two phonemes in the word read by the examiner and to report the resulting nonword. (ii) Phoneme Blending: this task assesses the capacity to derive a phonemic pattern from distinct phonemic units. The examiner presented each of 20 words letter by letter and the child was requested to identify and report the resulting word. For both tasks, the scores refer to the total number of errors, converted into z-scores. This test was administered to the total sample of 266 children.

All the children were individually tested in a separate, quiet room at the Institute before starting intervention. All the tests were administered by trained psychologists.

### 2.3. Data Analysis

Data analysis involved a large set of children with dyslexia (N = 266, 84 girls and 182 boys, mean age = 9.96, SD = 1.79, range 7–15). First of all, each of the two dependent variables, reading accuracy and speed, was computed as the average of the z-scores expressing the children’s performance on text, word and nonword reading. Then, we selected five predictors on grounds of (i) theoretical relevance, (ii) coverage of a wide range of cognitive domains, and (iii) having a non-negative correlation with both reading accuracy and reading speed (the two predicted variables). Among the several predictors taken from the same test or test battery, we chose the measure or combination of measures that maximized those correlations, in order to maximize statistical power of the later GLM analyses. The five predictors were: IQ, phonological awareness (accuracy z-score), RAN speed (computed as speed z-score), visual search speed (computed as speed z-score) and memory (computed as the average z-score for letter span forward and backward). All variables, predictor and dependent, were expressed as z-scores. Fifty-five (out of 266) subjects had all seven scores (five predictors plus two predicted variables). This subsample had 28 girls and 27 boys, mean age = 9.36, SD = 1.69, range 7–14.

The critical expectation was that a single predictor should have an effect on reading performance that depends on the sum of the other predictors; the idea is that (e.g.,) memory performance should contribute to reading performance, with a positive slope (the better the memory performance, the better the reading one), but that such a slope would be steeper when, collectively, the other cognitive abilities are lower. These cognitive abilities were quantified by an aggregate score, the “other factors index” (OF), which was obtained by combining the z-scores of the other predictors. Thus, for instance, if memory was the predictor under scrutiny, we combined the remaining predictors, IQ, RAN, visual search and phonological awareness, into a single OF index. This combined score was a weighted sum of the four z-scores, with the weights being the correlation coefficients of the four z-scores with the predicted independent variable (either reading accuracy or reading speed). The weights were obtained exploiting the whole N = 266 dataset, albeit only the 55 subjects who had all five predictors entered the subsequent GLM (general linear model, JAMOVI, [71]) analyses. Each GLM had the reading variable as Y, and (i) the single predictor, (ii) the OF index, and (iii) the predictor-by-OF-index interaction, as Xs. The predicted pattern was that of a negative interaction coefficient, corresponding to an amplification of the effect by the singled-out predictor, the lower the OF index derived from the other four predictors.

## 3. Results

Ten GLM analyses were performed, five for each reading variable (reading accuracy, reading speed). In all ten GLM analyses, the interaction parameter turned out to be negative, and in five cases it was significantly so (one-tailed *p* < 0.05, Table 1). 

Figure 1 shows the interactive patterns in such five cases; in all of them, as expected, the lower the OF index (X axis), the larger the effect of the specific predictor (that is, the larger the gap between the two regression lines, yellow and blue, corresponding, respectively, to high and low levels of the predictor in the sample).

As in five cases the predicted interaction did not turn out to be significant, we wondered whether these negative results showed a dichotomous pattern, with some interactions being really present and others not, or rather just some (plausible) heterogeneity in the size of the interaction effects, with some of them being too small to be detected with a N = 55 sample. Post hoc power analysis was not an option due to its circularity [74], so we used Bayes factors [73] in order to be able to disentangle cases in which a non-significant *p* value constituted evidence in favor of the null hypothesis of no interaction, and against the alternative hypothesis of its presence (conventionally, *BF*_10_ < 0.333), or rather, the result of lack of evidence for either hypothesis (0.333 < *BF*_10_ < 3). The advantage of this Bayesian approach is that it gives a continuous figure of the relative weight of evidence for the model with the interaction term vs. the model without the interaction term; that is, the additive model.

Table 2 reports the Bayes factor values (*BF*_10_). All results with significant *p* values in Table 1 were confirmed, with different degrees of relative evidence in favor of the presence of an interaction, ranging from 3.773 to 29.579. Conventionally, *BF*_10_ values between 3 and 10 are considered “moderate” amounts of evidence, and values between 10 and 30 as “strong” evidence. All the five cases in which the *p* value fell short of significance had inconclusive *BF*_10_ values (conventionally, in the range between 0.333 and 3), meaning that we do not have convincing evidence either for the presence or for the absence of an interaction effect. So, further data and research are needed for these specific predictor-reading score combinations. 

## 4. Discussion

Following the hypothesis that reading ability is the product of the interplay within a large set of neuropsychological functions, all of which are involved—to different extents—in the reading process, the present study retrospectively investigated the interactions between each function and the sum of the remaining functions in a sample of children with DD. To this aim, all the functions for which age norms exist were expressed as z-scores, and these scores were further assigned a specific weight based on their correlation with reading ability (separately considering reading accuracy and reading speed). Data on full-scale IQ, verbal memory, phonological awareness, RAN and visual search were included in a comprehensive index (named the MFi-index) resulting from the sum of all the weighted z-scores. Then, each of the five factors was included in an analysis to investigate its interaction with the MFi-index, separately on each of the two components of reading (accuracy and speed). Of course, in order to avoid obvious statistical redundancy, the singled-out factor was excluded from the MFi-index, which was then renamed OF index (because it contains all functions “other” than the singled-out factor; so for instance, visual search accuracy was studied together with the sum of all weighted scores except for visual search). The factor-by-OF interaction was significant for five out of the 10 statistical tests, specifically for the interactions involving memory and IQ for reading accuracy, and the interactions involving RAN speed, phonological awareness and visual search speed for reading speed. A Bayesian analysis allowed us to classify the evidence in favor of the expected interaction as “moderate” in three cases (memory on reading accuracy, RAN and phonological awareness on reading speed), and “strong” in the other two (IQ on reading accuracy, visual search on reading speed). As to the remaining five tests, evidence was inconclusive rather than in favor of the absence of an interaction.

The presence of these significant interactions confirms that the effect of each variable on reading should not be considered as additive with respect to the sum of the other ones, and that the impact each single component may have on the whole reading process is not absolute, but it depends on the efficiency level of the other functions. In other terms, even a relatively less crucial function may have a great impact on the resulting level of overall reading ability when other, usually more crucial functions are collectively impaired. 

### 4.1. Theoretical Implications

The present results go exactly in the direction hypothesized by the multifactor model of DD presented in the present paper. We are aware that such data do not constitute conclusive evidence yet, due to the relatively small number of participants included in the analysis and to the retrospective nature of the study, this rested on data collected to respond to different experimental questions and therefore only included some of the relevant functions. To start with, the selected sample did not include any information about visual or auditory attention, and the only variable related to visual processing was visual search speed. Moreover, data on executive functions were not available for this sample. The database, though, included a rather representative set of functions from the most crucial domains: phonological, mnestic, visual and intellectual; and crucially, it included both of the variables that are regarded as the main predictors of reading [75]: phonological awareness and RAN. Clearly, the larger the set of functions included, the smaller the portion of unknown (but existing) factors and—presumably—the more precise the model’s prediction can be. Conversely though, the inclusion of too many functions (especially if mutually correlated) would introduce overfitting issues and make testing of the hypothesis less reliable. Ideally then, the goal of future studies could be that of identifying a reasonably large set of non-correlated functions, determining their relative weights (possibly the most challenging task) and collecting a reasonably large set of rigorously controlled data for testing their contribution to DD severity. Even if, as stated in the Introduction, the set of functions involved in reading is potentially very large, any factor that has not been considered in the literature is likely to play a very limited role in the severity of the disorder, and thus its relative weight could be so small that the overall sum of unknown factors might be negligible. Therefore, albeit still not optimal, this database can be considered as a sufficient approximation of the complexity of the reading process, good enough to be the benchmark for a preliminary test of the model. Further, more comprehensive studies on larger samples and with longitudinal data will provide more reliable and complete information about the possibility for the model to explain the complex etiological pathways to dyslexia. The present results are in line with the proposals of various types of MDMs, as proposed by [30,31,32,45].

Theoretical implications of this work, beyond the support of multifactor models of DD, concern the non-additivity of the contribution that each single function brings to the reading process, especially in clinical conditions when one or more of the functions are impaired or weak. The present data bring preliminary evidence that the role that each function exerts depends on the status of the other functions (the OF index in our data). The more impaired the other factors, the more relevant the contribution of a given factor will be. This can explain the results of studies (e.g., [3,53,76]) where cutoffs were employed to select cases “with or without” a given impairment, and apparently showing that some of the impairments were represented in a small percentage of the sample, concluding that these were rather irrelevant components. This kind of approach rests on categorical models of impairment, where each function provides an additive contribution to reading ability, and where a fixed set of functions (either a single-deficit or a multiple-deficit set) is expected to predict the final outcome. In the approach proposed in the present work, the contribution of each function to reading is free—and expected—to change from one individual to another, depending on their functional profile. 

### 4.2. Practical and Clinical Implications

The clinical and practical implications of the proposal have little relevance for diagnosis (although they might be more relevant for the purposes of early risk detection and empowerment), as diagnosis should simply be based on reading performance itself (see also [31,53], among others), but they are very relevant for intervention. If the mixture of strengths and deficits is different in every individual with DD, the specific goals and means for intervention should also differ from case to case. 


*However, does this mean that each intervention program should be built and individually tailored to account for the specific profile of each single dyslexic child?*


Probably the best answer after the proposal discussed above should be an affirmative one. However, for practical and methodological reasons, such an approach could be difficult to sustain in a clinical context. First of all, it would be rather costly in terms of planning, as long as the planning has to be done by the clinician and is not supported by automated algorithms that use data from the neuropsychological profile and AI to build a fully detailed, individualized program. Second, the absence of a shared, unitary framework could make the validation of the approach itself rather difficult, and would thus reopen the door to the colorful but little scientifically justified variety of treatments that were the rule in the past and were finally rejected as “non-evidence-based”.

For these reasons, the “dyslexia subtypes approach” could be a viable strategy to reconcile a rigorous methodology with the individualization required by a multifactor model perspective. Indeed, DD subtypes can be viewed as an intermediate position between a universal, one-size-fits-all approach to intervention and a fully individualized approach: a subtype is in fact a cluster of characteristics that are often found together, either due to genetic links or to functional bondages between related functions. In this perspective, subtypes may be seen as a first step in the direction of analyzing interactions between the various functions and subprocesses which, as said before, cannot be considered as single factors of the reading process but, more appropriately, as elements engaged in a complex interplay. 

One classification linking diagnosis and intervention in a clear and straightforward way, and very close to a multifactor perspective, is the subtyping proposed in the balance model of dyslexia [77,78]. The classification into P-types (where P stands for perceptual, after the preferred reading strategy based on visual analysis, careful and slow) and L-types (where L stands for linguistic, again describing the preferred reading strategy, based on fast but inaccurate guessing based on linguistic and contextual cues) was proposed about 40 years ago, and it clearly needs to be examined and justified in the framework of current knowledge about reading and dyslexia. The distinction between these two types of DD rests on a theory of reading acquisition that emphasizes the predominant role of each of the two hemispheres (the right one—prevalent for visual analysis—in the earlier stages, and the left one—performing more and more complex linguistic anticipations—later on) in the consolidation of reading ability during the first years of schooling. Indeed, the idea that the involvement of the right hemisphere in reading decreases after the first months of school and that the reading process becomes more and more lateralized to the left hemisphere has been largely confirmed in the experimental [79,80] as well as in the neuroimaging literature [21,79,81,82,83,84]. 

A further extension of the classification has been proposed in later years by various authors [85,86], suggesting the existence of a mixed type of dyslexia, or M-type, characterized by slow and inaccurate reading, with a similar proportion of time-consuming and substantive errors. This category would practically include all the children who could not be classified as either P- or L-types. In empirical studies, M-types have been found to be the most prevalent subtype (representing 40% to more than 60% of children with DD, followed by P-types and L-types [87,88,89,90].

The classification proposed in the balance model was shown to capture different neuropsychological profiles that do not simply correspond to the classical distinction between phonological and surface dyslexia. Such differences include: (i) different reading styles and strategies [91,92,93]; (ii) performance on cognitive tests (L-types perform better than P-types on certain linguistic tasks, and worse than P-types on visuospatial tasks [91,92,93,94], but show inhibitory deficits and fronto-central dysfunctions [92,95,96]; (iii) morphology, lateral distribution and development of the lateral distribution of certain components of word-elicited brain responses [77,78,97,98]; (iv) differences in callosal functions (L-types, but not P-types, seem to have a deficit in the callosal transmission of tactile information [99], that could also be led back to a reduced ability to perform cognitive tasks requiring spatial representation in mirror-reversed conditions [100,101,102]). 

The composite profiles of L-, P- and M-types are thus to be considered not as separated subgroups but rather as more probable combinations of strengths and weaknesses, reflecting the descriptions of cognitive subtypes of DD as found in large-sample studies (e.g., [103,104,105]).


*Why should subtyping be viewed as a multifactor approach to intervention?*


In the original version of VHSS (visual hemisphere-specific stimulation) training, the aim of tachistoscopic presentation was essentially that of stimulating the hemisphere contralateral to the hemifield where the stimuli were presented. However, tachistoscopic presentation of words also requires fast and efficient activation of reading-related processes, including the various components of visual analysis, letter-to-sound mapping and phonological processes, verbal memory, manipulation of information and intra/inter-hemispheric transfer [88,90,106]. For this reason, in current versions of VHSS (see [90,107]) the variety of stimuli and tasks has been largely expanded so as to accommodate all the goals that training can be required to fulfil in a multifactor model, both at the perceptual and at the cognitive and strategic level.

The part of treatment devoted to L-types (and to M-types in the initial stages of treatment) emphasizes the visual–perceptual aspects of the stimuli, so as to train visual–perceptual and visual–attentional components of reading, from visual analysis of basic features to higher-level visual and attentional processes (e.g., selective attention, visual attentional span, crowding control, serial search, etc. [13,108,109,110,111,112,113,114]). Notably, it is in this part also that phonological awareness is mainly addressed (and indeed, nonwords are provided as stimuli). The second part of treatment, devoted to P-types (and to M-types in the more advanced phases of treatment), addresses linguistic processes that can be used for effective anticipation of the stimuli. Indeed, anticipation can occur based on semantic content, but also less obviously but not less crucially, through lexical, phonological, morphological and syntactic cues, insofar as specific sections of the word are progressively decoded, restricting the range of possible final candidates based on vocabulary knowledge, grammatical and phonotactic constraints [115]. 

Although the role of morphological competence in reading has been clearly shown in several studies, [116,117,118,119,120], it is usually described as either a compensatory ability or a protective factor, whereas it could be viewed as one of the skills contributing to reading (as it is often highlighted in the literature about dyslexia in Chinese). As to semantic and syntactic abilities, they are usually considered as being unaltered in dyslexic individuals [46], although finer-grained investigation methods can highlight subtle but evident deficits in these skills as well [121,122,123,124,125]. In this case also, evidence of impairments in linguistic skills is easily available, but it is generally interpreted as either resulting from the association of DD with language impairments or as a consequence of the phonological impairment—in both cases, then, as a causally unrelated factor (see, for instance, [126,127]). This interpretation is questionable, though, considering that the impairments are confirmed also when the presence of developmental language disorder and/or the impact of phonological impairment are controlled for (e.g., [124,128]). 

Whether or not the linguistic deficits in DD are a consequence of phonological impairments, it is certainly true that they exert an impact on reading efficiency and, if stimulated, they may take part in compensatory strategies [58,59,129,130,131]. Recent versions of the balance model can tap all aspects of linguistic anticipation, based on phonological, lexical, morphological, syntactic and semantic cues embedded in either highly or weakly constrained contexts, so as to ensure different levels of difficulty and a progression in therapeutic goals and processing demands. A more recent development is the combination of visual hemisphere-specific stimulation with an additional component of treatment addressing visual–spatial attention and control of crowding effects [109,111,114], specifically designed for remote intervention [107]. Moreover, it adds auditory input that was not included in previous treatment, thus allowing for the stimulation and empowerment of auditory-related functions. 

For all these reasons, we believe that a subtype-based approach to intervention such as (but certainly not only) the one described in the balance model very well fits the needs of a multifactor view of DD.

### 4.3. Further Directions and Developments

As anticipated, the empirical test of the hypothesis described in the present study is just a preliminary example of how a MFi-M can explain variation in the relationship between predictors of reading (based on neuropsychological reading models) and the degree of reading impairment in individuals with DD. Being a retrospective study, the five predictors used for this analysis do not represent the best possible set of predictors. Other functions such as auditory processing, attentional inhibition and executive functions are potential candidates for inclusion in the set. Other levels of the whole multifactor perspective, including the genetic, the social and the environmental levels, were not considered here but will certainly enrich the model once the exact picture of a single level (here, the cognitive–neuropsychological one) has been clearly defined.

Special consideration should be devoted to the issue of language specificity. Even if a common substrate is often described for individuals with DD across various languages and orthographic systems [132], it is widely recognized that different abilities play more crucial or more marginal roles depending on language variation [133,134]. Morphological awareness, for instance, has been identified as a very crucial ability for reading non-alphabetic languages such as Chinese [135]. Phonological awareness and RAN, on the other hand, have been found to be more crucial for alphabetic languages but their relative weight seems to differ in transparent versus opaque orthographic systems [136]. Therefore, it should be expected that, beyond individual variation, linguistic and cultural differences (including school systems and reading teaching methods) may also contribute in producing the different weights [137,138].

The final (but not last in priority) step would be the definition of a complete model for intervention. As previously argued, a subtype-based approach may constitute a good compromise between rigidly formulated protocols and completely individually tailored programs. Nonetheless, the relevant subtypes need to be more completely described in terms of all the relevant functions, and the corresponding treatments need to be enriched with modules and materials able to stimulate each of these functions (ideally taking into account the individual’s complete profile, as required by a MFi-M). We firmly believe that the support of technology and machine-learning algorithms will be extremely helpful to reach these goals in the near future. 

## 5. Conclusions

As a general conclusion of this preliminary validation, the MFi-M of DD seems to be able to fit empirical data that showed non-additive, interactive relationships between factors that predict dyslexia: in many cases, the impact that each factor has on the resulting reading ability depends on the collective degree of impairment (or strength) of the other factors. The set of factors to be included is still to be explored, and potentially ranges over all the neuropsychological functions that are known to take part in the various steps of the reading process. Each factor, moreover, has to be given a specific weight, that has to be determined taking into account the role it has been shown to be playing in typical and atypical reading activity.

As to the implications for intervention, a subtype-based approach combined with the choice of a neuropsychological framework that addresses the weaknesses and strengths of the various subtypes would be a reasonable compromise between one-size-fits-all approaches and completely individualized programs. The advantages of such a choice would be the ease of application, the presence of background validation data ensuring reliability, the evidence-based logic and the compatibility with a multifactor view of the disorder. 

## Figures and Tables

**Figure 1 brainsci-13-00328-f001:**
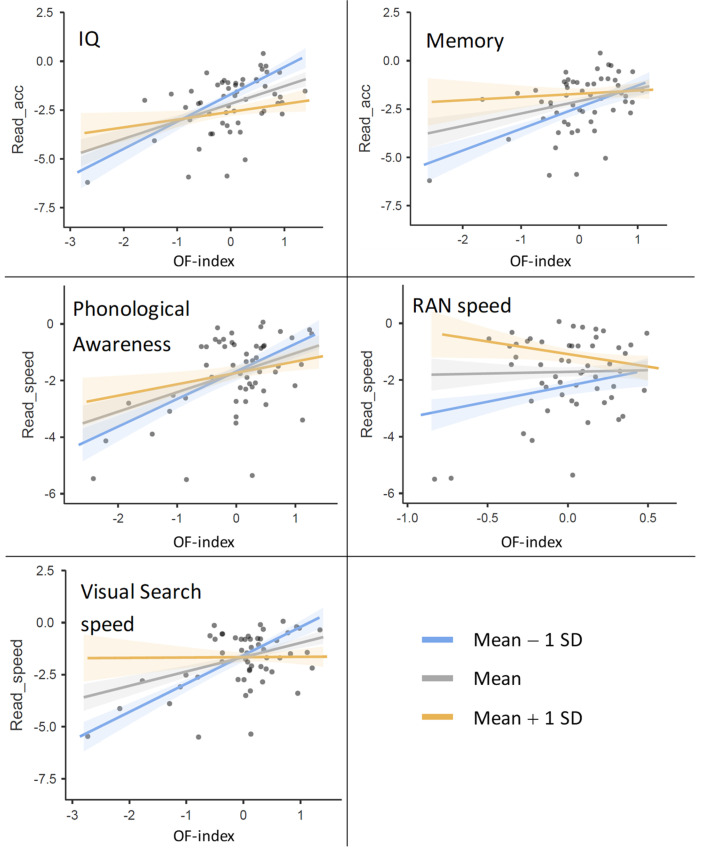
Each plot shows the interaction between one specific predictor (named in the top-left corner of the plot) and other predictors in determining reading performance. The latter was measured in terms of either reading accuracy (“acc”) or reading speed, and is reported as a z-score on the vertical axis. Dots represent single participants. The specific predictor varies across colors (different regression lines in the plot), while the other predictors are shown as a composite score, the “other factors” (OF) index, reported on the horizontal axis (see text for computational details). For instance, the first plot reports the effect by predictors other than IQ as the slope of the regression lines, and the separate regression lines show different levels of IQ: the gray line shows performance by subjects whose IQ was at the sample mean, the blue line shows performance by subjects whose IQ was 1 SD below the sample mean (−1 SD), and the yellow line shows performance by subjects whose IQ was 1 SD above the sample mean (+1 SD). Shaded regions around the regression lines show the standard error of the mean [73]. In all cases, the effect of the single predictor (visible as the gap between the yellow and the blue line) is significantly amplified when other cognitive performances are low (low OF index, on the X axis).

**Table 1 brainsci-13-00328-t001:** Interactions between specific predictors and OF index (the other factors combined) on reading scores.

		Interaction Predictor X OF Index		
Dependent Variable	Predictor	Estimate (SE)	t(51)	One-Tailed *p*
Reading accuracy	IQ	−0.611 (0.277)	−2.21	0.016
	Memory	−1.068 (0.58)	−1.84	0.036
	RAN speed	−0.277 (0.199)	−1.388	0.086
	Visual search speed	−0.0832 (0.239)	−0.349	0.365
	Phonological awareness	−0.13 (0.127)	−1.025	0.155
Reading speed	IQ	−0.293 (0.238)	−1.231	0.112
	Memory	−0.457 (0.49)	−0.931	0.178
	RAN speed	−0.706 (0.332)	−2.13	0.019
	Visual search speed	−0.5298 (0.184)	−2.882	0.003
	Phonological awareness	−0.227 (0.116)	−1.965	0.028

All *p* values are one-tailed in the expected direction, i.e., an amplification of the effect of the predictor (its positive slope) with decreasing OF index. Such a direction corresponds to a negative sign of the estimate and of the *t* value. No multiple-comparison correction was envisaged, because the interaction was predicted in all cases, and not in *at least one case* [72]. All *t* values have 51 degrees of freedom. SE: standard error of the estimate.

**Table 2 brainsci-13-00328-t002:** Bayes factors for the interaction terms between specific predictors and OF index (the other factors combined) on reading scores.

		Interaction Predictor × OF Index		
Dependent Variable	Predictor	Estimate (Adj. SE)	H_1_ Prior	BF_10_
Reading accuracy	IQ	−0.611 (0.279)	[−0.57, 0]	12.896
	Memory	−1.068 (0.584)	[−0.82, 0]	7.612
	RAN speed	−0.277 (0.201)	[−0.70, 0]	1.705
	Visual search speed	−0.0832 (0.241)	[−0.89, 0]	0.457
	Phonological awareness	−0.13 (0.128)	[−0.58, 0]	0.783
Reading speed	IQ	−0.293 (0.24)	[−0.68, 0]	1.658
	Memory	−0.457 (0.494)	[−0.78, 0]	1.972
	RAN speed	−0.706 (0.335)	[−0.97, 0]	7.868
	Visual search speed	−0.5298 (0.185)	[−0.93, 0]	29.579
	Phonological awareness	−0.227 (0.117)	[−0.50, 0]	3.773

*BF*_10_, Bayes factors obtained by the Dienes calculator (https://medstats.github.io/bayesfactor.html (accessed on 8 February 2023)). *SE*, standard errors, were adjusted by the term [1 + 20/(*df*)^2^] = 1.008, as indicated in [74], p. 11. *H*_1_
*Prior*: uniform prior distributions were used for the alternative hypothesis *H*_1_, in the reported ranges. See Appendix A for details on the way the ranges were computed.

## Data Availability

The data presented in this study have been deposited in Zenodo, https://doi.org/10.5281/zenodo.7618344. Access is possible upon written request to the corresponding author. The data are not publicly available due to ethical restrictions, and can be shared only under appropriate agreements.

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
