# Peer review of "Revisiting Multifactor Models of Dyslexia: Do They Fit Empirical Data and What Are Their Implications for Intervention?"

_brainsci, 2023, doi:10.3390/brainsci13020328_

Round 1
Reviewer 1 Report
I’ve read the paper twice and really enjoyed it! I think the writing in the introduction borders on brilliant, and I found much of the discussion very much worth reading, and truly thought provoking and informative. The major problem with the paper is the weakness of the study presented. It does not have the statistical power required for the research questions/hypotheses. I felt like the authors themselves are not confident in their own study. The interactions observed are not unpacked properly, nor are explained in the level of detail required. I also have a strong suspicion that the way the analyses are conducted is contradictory to the theoretical model being tested. You cannot collapse across a set of experimental measures, when a main assumption of the model is that there are important (continuum based) individual differences in those variables. If nothing else the OF analysis would obscure important variance and many more key interactions than the ones reported.
I’m torn on what to recommend for this paper, part of me thinks that it should be rejected on the basis of the empirical weaknesses in the study, and part of me thinks that this work should be available on the merits of the introduction and discussion. I would like to point out that the majority of discussion is unrelated (or not directly related) to the issues and hypotheses outlined in the introduction. The discussion contains one page relating to the current study and over 4 pages about subtypes and intervention implications.
Here is my recommendation:
1. Either eliminate the empirical study, and transform this into an editorial/review/commentary/perspective piece, retaining the main parts of the intro and discussion.
2. Strength and re-consider the statistical analyses of the study, and remove the 4+ pages of the discussion that is about subtypes and intervention. Then expand the true discussion of the actual results presented.
In short, there is a lot to like here, but at present, it does not work as an empirical study.
Reviewer 2 Report
The present study investigates the predictive value of 5 interacting factors in a multi-factor theory of developmental dyslexia based on a restrospective analysis of 55 children. The authors argue that interactions between factors play a significant role in predicting performance in support of the multi-factor model.
Major comments:
1. Clarity of description of the current analysis
Data analysis in section 2.2 (which should also be numbered 2.3) is difficult to follow. The predicted variables of Reading Accuracy and Reading Speed are not defined when they are introduced (line 300). Instead they are defined later (line 306). Either define them immediately, or introduce these composite measures of reading performance earlier (in 2.2 Materials after the two reading tests are explained at the bottom of page 5). My recommendation, given the current organization, is to define them in 2.3 before moving on to the predictor variables.
The IQ test as a predictor variable is also introduced here in 2.3 rather than as one of the materials in 2.2. I suggest explaining the IQ test as part of the materials in 2.2. I would also recommend using consistent labels from 2.2 in 2.3, so that, for example, "memory" doesn't switch from "verbal memory" in 2.2. to "memory" in 2.3 to "memory score" in table 1. Also, use variables in a consistent order. They are given in a different order in 2.2 versus in 2.3 (line 318) and then in the table.
The authors use a somewhat unusual method to investigate interactions in their data. Typically, individual predictor variables can have an interaction term in a GLM and this might be what is expected. Since there are 5 variables here, the possiblity of 3, 4, and 5 way interaction in the standard GLM would be pretty impossible to interpret, so I think the authors have taken a reasonable approach. However, they should explicitly address this, and since they have run multiple models they should also explicitly address whether significance was adjusted for multiple tests (e.g. Bernoulli correction) or not.
2. Model comparison/discussion/conclusions
The authors did not actually test a linear/additive model (which is argued against in the discussion). They could easily do this with the existing data, I believe. So what are the results if the five predictors are used in a model with no interaction terms to predict reading accuracy and reading speed? In that model, which factors are significant predictors? How do the different models compare in accounting for the overall data (e.g. in R^2 for example, amount of variance described)?
Minor comments:
Page 2 line 68, perhaps adherents rather than "adhesions". And trim references 30-35 to remove multiple papers by same authors (i.e. 30 or 31, 34 or 35) to better represent the balance?
Page 2 line 84, perhaps intervention rather than "prevention" and removing the next sentence. I think the idea of prevention of dyslexia would require a lot more explaining if that's what the authors really mean to say. (which it looks like the authors get to two paragraphs below as they introduce the idea of protective factors in interactions) - This comment might not be relevant if the authors accept the recommendation in the next comment as the subsection heading would make it clearer that explanation is coming.
Page 2 line 87 (and later line 160) perhaps these single sentence paragraphs could be converted to subsection headings to more explicitly show the organization of ideas. As a reader, I eventually figured out these were topic transitions.
Page 5 line 209, is "former factor" the "single deficit" mentioned earlier in the paragraph. If so, I think that's a better wording, as I am unsure of the referent meant by "former factor"
Page 5 line 225, add "that met the inclusion criteria". As it reads now, the participants paragraph leaves open the possibility that only a subset of the available data were used which might lead the reader to suspect cherry picking of the data. For proper scientific rigor, this study must have considered all the available data that meet inclusion criteria.
Page 5 line 227, "in the past years" adds no information. Either delete or give the year range.
Page 5 line 252, change "the last grade of junior high school" to a numeric description, e.g. 8th grade as the reader does not know the number of years of schooling in the range unless they are familiar with the system in Italy.
Page 6 line 265-279 and later, I think something was inadvertently deleted after "RAN task (Rapid Automatized", and also the next task (starting line 272) is also labeled "RAN" so these two bullet points need to be corrected and clarified. If two different speeded naming tasks were used, this will need to be better explained in the analysis section (line 205 where a single RAN speed is given).
Page 7 line 310, explicitly include "and 27 boys" for two reasons. 1) why make the reader do the math?, and 2) it then becomes clear that the age data applies to the entire sample and not just the girls.
Round 2
Reviewer 1 Report
I think the authors have addressed my comments sufficiently well. The dataset is not ideal for a strong test of the theoretical issues. however, the authors have clearly addressed this point with their additions to the discussion. As I said in my previous review, there is a lot to like about the work, and in the end, the positives outweigh the reservations I had/have about the empirical data. The Bayes additions also help to address my reservations.